# The insulin sensitivity Mcauley index (MCAi) is associated with 40-year cancer mortality in a cohort of men and women free of diabetes at baseline

**Yonatan Moshkovits**[1,2], **David Rott**[2], **Angela Chetrit**[3], **Rachel Dankner**[1,3]*

**1** Department of Epidemiology and Preventive Medicine, School of Public Health, Sackler School of Medicine, Tel Aviv University, Tel Aviv-Yafo, Israel, **2** Leviev Heart Center, Sheba Medical Center, Ramat Gan, Israel, **3** Unit for Cardiovascular Epidemiology, the Gertner Institute for Epidemiology and Health Policy Research, Sheba Medical Center, Ramat Gan, Israel

* racheld@gertner.health.gov.il

**Data Availability Statement:** Data cannot be shared publicly because providing full patient-level information can potentially lead to their identification and compromise their confidentiality.

## Abstract

### Background

The association between insulin resistance and cancer-mortality is not fully explored. We investigated the association between several insulin sensitivity indices (ISIs) and cancer-mortality over 3.5 decades in a cohort of adult men and women. We hypothesized that higher insulin resistance will be associated with greater cancer-mortality risk.

### Methods

A cohort of 1,612 men and women free of diabetes during baseline were followed since 1979 through 2016 according to level of insulin resistance (IR) for cause specific mortality, as part of the Israel study on Glucose Intolerance, Obesity and Hypertension (GOH). IR was defined according to the Mcauley index (MCAi), calculated by fasting insulin and triglycerides, the Homeostatic Model Assessment (HOMA), the Matsuda Insulin Sensitivity Index (MISI), and the Quantitative Insulin Sensitivity Check Index (QUICKI), calculated by plasma glucose and insulin.

### Results

Mean age at baseline was 51.5 ± 8.0 years, 804 (49.9%) were males and 871 (54.0%) had prediabetes. Mean follow-up was 36.7±0.2 years and 47,191 person years were accrued. Cox proportional hazard model and competing risks analysis adjusted for age, sex, country of origin, BMI, blood pressure, total cholesterol, smoking and glycemic status, revealed an increased risk for cancer-mortality, HR = 1.5 (95% CI: 1.1–2.0, p = 0.005) for the MCAi $Q_1$ compared with $Q_{2-4}$. No statistically significant associations were observed between the other ISIs and cancer-mortality.

Requests for aggregated (anonymized) data related to this study can be obtained via Ms. Arnona Ziv, Director of the Gertner Institute Unit for Data Management and Computerization (contact via email: ArnonaZ@gertner.health.gov.il; Phone: +972-3-7731500) for researchers who meet the criteria for access to confidential data.

**Funding:** The author(s) received no specific funding for this work.

**Competing interests:** The authors have declared that no competing interests exist.

## Conclusion

The MCAi was independently associated with an increased risk for cancer-mortality in adult men and women free of diabetes and should be further studied as an early biomarker for cancer risk.

## 1. Introduction

Cancer remains a leading cause for morbidity and mortality in the US and worldwide [1, 2], with 9.5 million cancer-related deaths reported in 2018 in the world [3]. In Israel, cancer is the leading cause of death, with 13,050 deaths (25.4% of all deaths) reported in 2018 [4]. A number of factors were associated with cancer risk such as smoking [5], Body Mass Index [6], diabetes [7] and sedentary lifestyle [8].

The association between insulin resistance (IR) and cancer remain unclear. Metabolic alterations were previously found to correlate with both IR and cancer through dietary risk factors (e.g. hypercaloric diet, low fibers etc.) that induces inflammation and oxidative stress, or promote IGF-1 secretion that acts as a strong mitogen [9]. The common soil hypothesis suggest that in susceptible individuals, metabolic abnormalities such as obesity, IR and dyslipidemia would be the initial manifestation of unhealthy diets and lifestyle, whereas carcinogenesis is more prolonged with delayed clinical manifestations [10]. A wide epidemiologic evidence is showing that diabetes is strongly associated with specific types of cancer [6], mainly pancreatic and liver cancer [11]. The American Diabetes Association and the American Cancer Society issued a consensus report on diabetes association with cancer incidence [12]. Nevertheless, the nature of this association is yet to be clarified, with the possibility of an indirect association, underlined by the hyperinsulinemic state characteristic of newly diagnosed individuals with diabetes, or by glucose lowering medication use, in addition to shared risk factors such as obesity [13–15].

The associations between type 2 diabetes, IR and increased fasting glucose plasma levels with malignancy associated mortality were demonstrated in a number of studies [15, 16], however these studies were mostly on diabetic participants or with a short follow-up period. Furthermore other studies on diabetic patients, including meta-analyses, did not demonstrate such an association with cancer mortality [17].

Insulin resistance can be evaluated indirectly via validated indices, calculated using insulin and glucose blood levels [18–21]. Frequently examined indices include the Homeostatic Model Assessment (HOMA) [21], the Matsuda Insulin Sensitivity Index (MISI) [18], the Quantitative Insulin Sensitivity Check Index (QUICKI) [20] and the Mcauley index (MCAi) [19]. Albeit values of insulin sensitivity indices (ISIs), denoting IR, were associated with an increased risk for specific types of cancer, e.g., prostate [22] and endometrial cancer [23], such an association with malignancy associated mortality is still not established, with contradicting results [16, 24, 25].

We aimed to investigate the association between IR surrogates, i.e., fasting insulin, fasting plasma glucose levels and several ISIs, with cancer mortality, in a cohort of adult healthy men and women over a 40-year follow-up.

## 2. Materials and methods

### 2.1 Study design and population

This is a prospective cohort study of adult men and women, who were randomly drawn from the national population registry in 1967 according to strata of sex, country of birth to establish the 4 main Jewish ethnic groups (Yemenite, Asian, North Africans, and European-North Americans) and birth decade (1912–1921; 1922–1931; 1932–1941). In the second phase of the Israel study on Glucose intolerance, Obesity, and Hypertension (GOH), performed between

1979 and 1982, participants were measured for weight, height, and blood pressure, and gave blood after a12-hour fast for glucose and insulin. They also did a 2-hour oral glucose tolerance test (OGTT) after an ingestion of 100 gr glucose.

Inclusion criteria for the current study included the absence of diabetes at baseline and the availability of data on both fasting glucose and insulin plasma levels at baseline. Individuals who died from cancer within the first 2 years of follow-up were excluded from the cohort. Out of 2,769 participants primarily examined in the second phase, 1,612 met the inclusion criteria.

The final sample showed similar characteristics as the original cohort in age, sex, ethnicity, blood pressure, and Body Mass index (BMI) distribution. Information regarding the GOH population and methodology was previously detailed elsewhere [24, 26].

Blood glucose was measured using the automated Technicon Autoanalyser II (Technicon Instruments Corp, Tarrytown, NY); Blood insulin was measured using the Phadebas Radioimmunoassay kit (Pharmacia Diagnostics Inc. Piscataway, NJ). Blood test analysis was performed by a single laboratory.

Participants were followed until December 2016 for malignancy associated mortality. Participant's approval was obtained a priori by their verbal free consent to participate in medical interviews and blood tests, and the study protocol was approved by the Sheba Medical Center's IRB.

## 2.2 Insulin resistance

The current study examined IR state as reflected by the following IR surrogates and ISIs: Fasting insulin and glucose plasma levels: both were categorized into quartiles and the upper quartile ($Q_4$) was compared to the lower quartiles ($Q_{1-3}$) as with the ISIs.

ISIs were calculated as follow [18–21, 27]:

Homeostatic model assessment (HOMA)-Insulin resistance (IR) and beta cell function (% B) [21], were calculated as follows:

$$HOMA1 - IR = FPI \times \frac{FPG}{405}$$

$$HOMA1 - \%B = 360 \times \frac{FPI}{(FPG - 63)}$$

Matsuda Insulin Sensitivity Index (MISI) [18]:

$$MISI = \frac{10,000}{\sqrt{(FPG \times FPI) \times [\text{mean glucose during OGTT x mean insulin during OGTT}]}}$$

MISI mean glucose and mean insulin were calculated using glucose taken at *0*, 60 and 120 minutes during the OGTT.

Quantitative Insulin Sensitivity Check Index (QUICKI) [20] was calculated as follows:

$$QUICKI = \frac{1}{\text{Log Fasting Plasma Insulin} + \text{Log Fasting Plasma Glucose}}$$

Mcauley index (MCAi) [19] was calculated as follows:

$MCAi = e^{[2.63 - 0.28*Ln\ FPI - 0.31*Ln\ trig]}$ Were FPI refer to fasting insulin levels in $\left[\frac{mU}{L}\right]$; FPG refer to fasting glucose levels in $\left[\frac{Mg}{dl}\right]$; Trig refer to fasting triglycerides levels in $\left[\frac{mMole}{L}\right]$.

ISIs that were not normally distributed were logarithmically transformed using natural log (ln). Lower values of HOMA-%B, MISI, QUICKI and MCAi, and higher values of HOMA-IR depicts insulin resistance.

ISIs characteristics, description and classification are further elaborated elsewhere [28].

## 2.3 Death from cancer

The primary outcome was the 40-year mortality rate due to malignancy, reported as the primary cause of death, using the International Classification of Diseases (ICD) 9 or ICD 10. Mortality date was obtained from the Israeli population registry through December 2016. Follow-up started at the baseline examination date and ended at date of death or by the end of the follow-up, whichever occurred first.

## 2.4. Statistical methods

Sample size was calculated using WINPEPI software implementing the Z test for proportion analysis with 80% power and 5% significance level. Based on previous publication on the study cohort [29], assuming an average probability of survival at the end of follow-up of 35% for individuals at the higher quartiles of the ISIs and a minimal Hazard ratio of 1.2, the necessary total sample size was 1,548 subjects.

Chi square test or Fisher's exact test for small cells were performed in order to evaluate differences among ISI quartiles. One-way ANOVA test for normally distributed variables or the Kruskal Wallis test for nonparametric variables were used for continuous variables, with two-sided p-values (p) set at the 0.05 level of significance in order to evaluate differences between those who remained alive, those who died from cancer and those who died from other causes by the end of the follow-up. The associations between ISIs and 40-year cancer death rate were examined for cumulative incidence analysis using the Cox proportional hazards model. Study participants were censored at the time of non-cancer deaths or by the end of follow-up, whichever came first. An additional approach used death from non-cancer causes as a competing risk to cancer death (the Fine and Gray method [30]) by calculating the sub-distribution hazard ratio (SHR). This method is based on the Cumulative Incidence Function (CIF) that counts failures from competing events and deaths from the primary endpoint, whereas the competing events in the cumulative incidence method are censored. Each insulin resistance surrogate was evaluated in a separate model. In order to avoid multicollinearity, Spearman's rank correlation coefficient test was performed, excluding covariates with a correlation of 60% or above from the same model. Survival analysis was performed according to cause specific mortality (i.e. deaths from cancer vs survival and non-cancer deaths). Models were adjusted for demographic variables as for known mortality risk factors such as smoking status, systolic blood pressure, BMI, cholesterol and glycemic status. Models are presented with Hazard Ratio (HR) or SHR with 95% confidence intervals (95%CI). The proportional hazards assumption was tested using the log minus log plot and by constructing an interaction variable composed of time-to-event multiplied by the covariate and adding it into the model.

Kaplan Meier survival curves for IR surrogates were compared using the log-rank test. Stratified analysis was conducted by glycemic status (i.e normoglycemia and prediabetes). In addition, we examined for an interaction between ISIs and sex.

Statistical analysis was performed using SPSS version 25.0.

## 3. Results

### 3.1. Baseline characteristics

A total of 1,612 subjects were followed until December 2016 for malignancy associated mortality. Table 1 presents the cohort baseline characteristics according to survival status and cause

**Table 1. Characteristics of 1,612 men and women free of diabetes at baseline (1979) according to vital status by the end of follow-up (2016).**

| Baseline characteristic | Total N (%) | Vital status by end of follow-up* | | | P-value |
|---|---|---|---|---|---|
| | | Alive Mean ± SD | Cancer death Mean ± SD | Non-cancer death Mean ± SD | |
| Number | 1612 | 642 | 264 | 706 | |
| Age (years), mean ± SD | 51.4 ± 8.0 | 46.3 ± 5.9 | 53.3 ± 7.6 | 55.3 ± 7.2 | <0.001 |
| Sex | | | | | <0.001 |
| Male | 804 (49.9) | 274 (42.7) | 149 (56.4) | 381 (54.0) | |
| Female | 808 (50.1) | 368 (57.3) | 115 (43.6) | 325 (46.0) | |
| Origin | | | | | 0.598 |
| Middle East | 428 (26.6) | 172 (26.8) | 69 (26.1) | 187 (23.7) | |
| North Africa | 288 (17.9) | 111 (17.3) | 48 (18.2) | 129 (18.3) | |
| Yemen | 351 (21.8) | 134 (20.9) | 50 (18.9) | 167 (23.7) | |
| Europe-America | 545 (33.8) | 225 (35.0) | 97 (36.7) | 223 (31.6) | |
| Smoking status [a] | | | | | 0.051 |
| Ever smoked | 634 (39.3) | 234 (36.4) | 119 (45.1) | 281 (39.8) | |
| Never-Smoker | 978 (60.7) | 408 (63.6) | 145 (54.9) | 425 (60.2) | |
| Glycemic state | | | | | <0.001 |
| Normoglycemia | 741 (46.0) | 346 (53.9) | 116 (43.9) | 279 (39.5) | |
| Prediabetes | 871 (54.0) | 296 (46.1) | 148 (56.1) | 427 (60.5) | |
| Blood Pressure (mmHg), mean ± SD | | | | | |
| Systolic | 130.4 ± 25.7 | 122.3 ± 23.6 | 132.8 ± 25.1 | 136.8 ± 25.7 | <0.001 |
| Diastolic | 83.4 ± 14.8 | 80.6 ± 15.2 | 83.5 ± 12.9 | 85.9 ± 15.2 | <0.001 |
| BMI (Kg/m$^2$) [b], median [IQR] | 25.3 [4.9] | 24.8 [4.1] | 25.3 [5.0] | 26.0 [5.3] | <0.001 |
| Normal | 735 (45.6) | 333 (51.9) | 121 (45.8) | 281 (39.8) | <0.001 |
| Overweight | 650 (40.3) | 248 (38.6) | 109 (41.3) | 293 (41.5) | |
| Obese | 227 (14.1) | 61 (9.5) | 34 (12.9) | 132 (18.7) | |
| Fasting glucose (mg/dl) | 97.8 ± 10.2 | 96.3 ± 9.7 | 98.4 ±10.8 | 98.9 ± 10.3 | <0.001 |
| $Q_{1-3}$ | 1108 (68.7) | 472 (73.5) | 175 (66.3) | 461 (65.3) | 0.003 |
| $Q_4$ | 504 (31.3) | 170 (26.5) | 89 (33.7) | 245 (34.7) | |
| Fasting insulin (mU/L) | 15.5 ± 10.4 | 14.9 ± 10.6 | 15.5 ± 8.8 | 15.9 ± 10.8 | 0.001 |
| $Q_{1-3}$ | 1191 (73.9) | 491 (76.5) | 189 (71.6) | 511 (72.4) | 0.150 |
| $Q_4$ | 421 (26.1) | 151 (23.5) | 75 (28.4) | 195 (27.6) | |
| Total cholesterol [c] (mg/dl), mean ± SD | 220.5 ± 54.1 | 216.3 ± 52.5 | 216.9 ± 54.5 | 225.7 ± 55.1 | 0.003 |
| Normal | 525 (32.7) | 222 (34.6) | 96 (36.4) | 209 (29.6) | 0.058 |
| Borderline-high | 501 (31.1) | 208 (32.4) | 77 (29.2) | 216 (30.6) | |
| High | 584 (36.2) | 212 (33.0) | 91 (34.5) | 281 (39.8) | |
| Triglycerides (mg/dl), median [IQR] | 110 [70] | 100 [70] | 120 [75] | 115 [75] | <0.001 |
| MISI †, median [IQR] | 3.6 [2.3] | 3.5 [2.2] | 2.9 [2.1] | 3.2 [2.2] | 0.320 |
| Ln MISI, mean ± SD | 1. 3 ± 0.5 | 1.2 ± 0.5 | 1.1 ± 0.6 | 1.1 ± 0.5 | 0.426 |
| $Q_1$ | 273 (27.7) | 87 (21.0) | 47 (28.8) | 112 (27.5) | 0.176 |
| $Q_{2-4}$ | 709 (72.1) | 327 (79.0) | 116 (71.2) | 295 (72.5) | |
| HOMA-IR ‡, Median [IQR] | 3.1 [2.1] | 3.0 [2.0] | 3.1 [2.2] | 3.3 [3] | 0.016 |
| Ln HOMA-IR, mean ± SD | 1.1 ± 0.6 | 1.1 ± 0.5 | 1.2 ± 0.6 | 1.2 ± 0.6 | 0.037 |
| $Q_{1-3}$ | 1209 (75) | 502 (78.2) | 192 (72.7) | 515 (72.9) | 0.055 |
| $Q_4$ | 403 (25) | 140 (21.8) | 72 (27.3) | 191 (27.1) | |
| HOMA-%B §, Median [IQR] | 142.3 [100.5] | 143.3 [105.0] | 145.9 [110.2] | 140.1 [95] | 0.735 |
| Ln HOMA-%B, mean ± SD | 4.9 ± 0.6 | 5.0 ± 0.6 | 4.9 ± 0.6 | 4.9 ± 0.6 | 0.693 |
| $Q_1$ | 402 (24.9) | 153 (23.8) | 74 (28.0) | 175 (24.8) | 0.413 |
| $Q_{2-4}$ | 1208 (74.9) | 489 (76.2) | 190 (72.0) | 529 (74.9) | |

(*Continued*)

**Table 1.** (Continued)

| Baseline characteristic | Total N (%) | Vital status by end of follow-up* | | | |
| --- | --- | --- | --- | --- | --- |
| | | Alive Mean ± SD | Cancer death Mean ± SD | Non-cancer death Mean ± SD | P-value |
| **Number** | **1612** | **642** | **264** | **706** | |
| QUICKI ɟ, mean ± SD | 0.32 ± 0.03 | 0.32 ± 0.02 | 0.32 ± 0.03 | 0.32 ± 0.03 | 0.091 |
| $Q_1$ | 401 (24.9) | 140 (21.8) | 71 (26.9) | 190 (26.9) | 0.063 |
| $Q_{2-4}$ | 1208 (74.9) | 502 (78.2) | 192 (72.7) | 514 (72.8) | |
| MCAi,¥ mean ± SD | 3.9 ± 0.9 | 4.0 ± 0.9 | 3.9 ± 0.9 | 3.9 ± 0.9 | 0.006 |
| $Q_1$ | 395 (24.5) | 137 (21.3) | 77 (29.2) | 181 (25.6) | 0.020 |
| $Q_{2-4}$ | 1187 (73.6) | 499 (77.7) | 182 (68.9) | 506 (71.7) | |

Between-group differences (alive vs cancer death vs non-cancer deaths) of categorical variables were examined using Chi square test or Fisher's exact test for small cells. Between-group differences of continuous variables were examined using student one-way Anova test for normally distributed variables or the Kruskal Wallis test for nonparametric variables, with two-sided p-values (p) set at 0.05 level of significance.

‡ HOMA-IR, Homeostatic model assessment -Insulin resistance; § HOMA-%B—Homeostatic model assessment–percent beta cell function

† MISI, Matsuda Insulin Sensitivity Index

ɟ QUICKI, Quantitative Insulin Sensitivity Check Index; ¥ MCAi, Mcauley index.

[a] Smoking status classification: Smoker-currently or past smoker. Nonsmoker-never smoked

[b] BMI categories: Normal $< 25$ kg/m$^2$, Overweight, 25–29.9 kg/m$^2$, Obese- BMI $\geq 30$ kg/m$^2$

[c] Total cholesterol classification: Normal $< 200$ mg/dl, Borderline-high, 200–239 mg/dl, High $\geq 240$ mg/dl.

of death. Mean age at baseline was 51.5 ± 8.0 years, 804 (49.9%) were males, 227 (14.1%) were obese (BMI $\geq 30$ kg/m$^2$) and 871 (54.0%) were had prediabetes.

## 3.2 Cancer related mortality

During a mean follow up time of 36.7±0.2 years, 970 (60.2%) participants died, and 47,191 person years were accrued. Cancer was the second most common cause of death with 264 (16.4%) deaths attributed to cancer related mortality.

Prediabetes was less prevalent among survivors, while cohort members who died from cancer were of male predominance, with higher rates of smoking, with increased blood pressure and BMI, and with higher fasting glucose, insulin and total triglycerides plasma levels. They were also more frequently found in the IR quartiles of ln MISI, ln HOMA-IR, ln HOMA-%B, QUICKI and MCAi.

Compared to individuals who died from cancer, those who died from other, non-cancer related, primarily cardiovascular causes, were older (P<0.001), with increased systolic (P = 0.031) and diastolic (P = 0.024) blood pressure, as well as higher total cholesterol (P = 0.028) (not shown).

S1 Table is presenting the distribution of site-specific cancer deaths. Almost 1/3 of all malignancies were of the digestive system (35.7%), followed by cancer of the genitourinary system, mostly prostate, then by lung cancer, non-solid tumors, and breast cancer.

The univariate analysis revealed that age, sex, smoking, hypertension, Ln-MISI and MCAi were significantly associated with cancer specific mortality (Table 2).

The adjusted multivariable analysis (for age, sex, origin, BMI, systolic blood pressure, cholesterol, smoking and glycemic status) revealed a significantly higher risk for cancer mortality for individuals in the MCAi $Q_1$, HR = 1.5 (95% CI: 1.1, 2.0, p = 0.005), as compared with the MCAi $Q_{2-4}$.

The other ISIs did not demonstrated such an association with cancer mortality. Age was independently associated with higher risk for cancer death in the adjusted model.

**Table 2. Cox regression models for associations between baseline characteristics of 1,610 men and women free of diabetes at baseline and cancer mortality over a mean follow-up of 36.7 years.**

| Characteristic | Reference category | Cancer mortality | | |
| --- | --- | --- | --- | --- |
| | | Univariate HR (95% CI) | Multivariate [a] HR (95% CI) | Competing risk Multivariate [b] SHR (95% CI) |
| Age | 10-year increment | 2.1 (1.8–2.5) | 2.1 (1.8–2.5) | 1.5 (1.3–1.7) |
| Sex, Male | Female | 1.5 (1.2–1.9) | 1.3 (0.97–1.7) | 1.2 (0.9–1.6) |
| Origin | Yemen | | | |
| Middle East | | 0.9 (0.6–1.2) | 0.9 (0.6–1.2) | 0.9 (0.7–1.3) |
| North Africa | | 0.9 (0.6–1.2) | 0.98 (0.7–1.3) | 0.99 (0.7–1.4) |
| Europe-America | | 0.96 (0.7–1.4) | 1.0 (0.7–1.5) | 0.8 (0.6–1.1) |
| Smoking status, Ever | Never | 1.3 (1.0–1.7) | 1.2 (0.9–1.6) | 1.2 (0.9–1.6) |
| Glycemic state | Normoglycemia | | | |
| Prediabetes | | 1.3 (0.98–1.6) | 1.0 (0.8–1.3) | 0.99 (0.8–1.3) |
| Systolic Blood Pressure | 1mmHg increment | 1.0 (1.0–1.01) | 1.0 (0.99–1.01) | 1.0 (0.99–1.01) |
| BMI (Kg/m$^2$) [c] | Normal | | | |
| Overweight | | 1.1 (0.8–1.4) | 0.9 (0.7–1.2) | 1.1 (0.8–1.7) |
| Obese | | 1.0 (0.7–1.5) | 0.8 (0.6–1.3) | 1.2 (0.8–1.8) |
| Total cholesterol [d] | Normal | | | |
| Borderline-high | | 0.8 (0.6–1.1) | 0.8 (0.5–1.0) | 0.8 (0.6–1.1) |
| High | | 0.9 (0.7–1.3) | 0.7 (0.5–0.9) | 0.7 (0.5–0.97) |
| Fasting triglycerides | 1 mg/dl increment | 1.0 (1.0–1.01) | 1.0 (0.99–1.0) | 1.0 (0.99–1.0) |
| MCAi [¥], Q$_1$ | Q$_{2-4}$ | 1.4 (1.1–1.8) | 1.5 (1.1–2.0) | 1.4 (1.1–1.9) |
| Ln MISI [†], Q$_1$ | Q$_{2-4}$ | 1.4 (1.0–1.9) | 1.3 (0.9–1.9) | 1.3 (0.9–2.0) |
| Ln HOMA-IR [‡], Q$_4$ | Q$_{1-3}$ | 1.2 (0.9–1.6) | 1.2 (0.9–1.6) | 1.2 (0.9–1.6) |
| Ln HOMA-%B [§], Q$_1$ | Q$_{2-4}$ | 1.2 (0.9–1.6) | 1.1 (0.8–1.5) | 1.1 (0.8–1.5) |
| QUICKI [¶], Q$_1$ | Q$_{2-4}$ | 1.2 (0.9–1.5) | 1.2 (0.9–1.5) | 1.2 (0.9–1.5) |
| Fasting insulin, Q$_4$ | Q$_{1-3}$ | 1.2 (0.9–1.5) | 1.2 (0.9–1.6) | 1.2 (0.9–1.6) |
| Fasting glucose, Q$_4$ | Q$_{1-3}$ | 1.2 (0.96–1.6) | 1.1 (0.8–1.4) | 1.1 (0.8–1.4) |

Adjusted covariates are reported using the final models that included the Mcauley index.

‡ HOMA-IR, Homeostatic model assessment -Insulin resistance

§ HOMA-%B—Homeostatic model assessment–percent beta cell function

† MISI, Matsuda Insulin Sensitivity Index

¶ QUICKI, Quantitative Insulin Sensitivity Check Index

¥ MCAi, Mcauley index.

[a] Multivariable models using cumulative incidence analysis/ cause specific mortality, comparing deaths from cancer with survivals and non-cancer deaths. The analyses were adjusted for: age, sex, origin, BMI, systolic blood pressure, cholesterol, smoking and diabetes status and to the MCAi and not the other insulin sensitivity indices

[b] Sub-distribution hazard ratio using death from non-cancer causes as competing risks (the Fine and Gray method)

[c] BMI categories: Normal < 25 kg/m$^2$, Overweight, 25–29.9 kg/m$^2$; Obese- BMI ≥ 30 kg/m$^2$

[d] Total cholesterol categories: Normal < 200 mg/dl; Borderline high 200–239 mg/dl; High ≥ 240 mg/dl.

In line with the cause specific mortality results, the Fine and Gray competing risks analysis revealed a significantly higher risk for cancer mortality for IR individuals in the MCAi Q$_1$, SHR = 1.4 (95% CI: 1.1, 1.9, P = 0.022). The remaining ISI surrogates did not show a significant association with cancer mortality, similar to the cause specific mortality results.

The Kaplan-Meier survival curves and log-rank test demonstrated a significant shorter times until cancer death for individuals in the IR MCAi quartile (Q$_1$) as compared to the upper quartiles, p = 0.02 (not shown).

Adjusted survival curves using Cox regression showed a significant shorter times until cancer death for individuals in the MCAi quartile (Q$_1$), p = 0.004 (Fig 1).

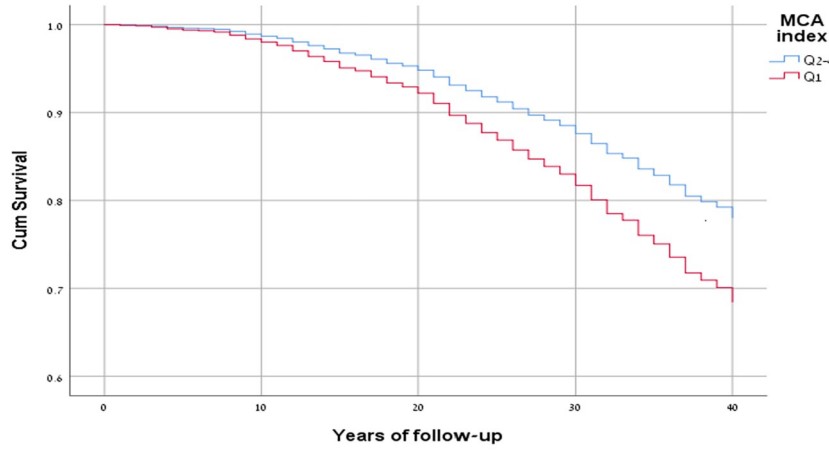

**Fig 1. Adjusted[a] survival curves using the Cox proportional hazard model, according to the Mcauley index low vs. higher quartiles for cancer mortality.** [a] Adjusted for: age, sex, origin, BMI, systolic blood pressure, cholesterol, smoking and diabetes status. Mean survival time for malignancy associated mortality in the lower (higher insulin resistance) MCA quartile ($Q_1$) was 35.8 (95%CI, 34.9–36.7) years and 36.9 (95%CI, 36.5–37.4) years in the upper (lower insulin resistance) MCA quartiles ($Q_{2-4}$), p = 0.02. Censoring occurred at time of other non-cancer death or end of follow-up.

An interaction between MCAi and glycemic state (i.e. normoglycemia vs prediabetes) was not found (p for interaction = 0.13).

Stratified analyses were conducted according to glycemic state. In the prediabetes group (n = 850), both cumulative incidence analysis using the Cox proportional hazards model and the competing risk analysis demonstrated an increased risk for cancer mortality for the MCAi $Q_1$, HR = 1.6 (95% CI: 1.1, 2.4, p = 0.013) and SHR = 1.7 (95% CI: 1.1, 2.5, p = 0.009), as compared with the MCAi $Q_{2-4}$. Such an association was not observed in the normoglycemia group (n = 741).

No interaction was observed between sex and the ISIs (p for interaction = 0.4, 0.1, 0.1, 0.2, 0.6 for MCAi, HOMA-IR, QUICKI, MISI and HOMA-%B respectively).

## 4. Discussion

In this long-term follow up of 1,612 men and women free of diabetes, over a mean period of 37 years, and close to 50,000 person-years, a significant association was demonstrated between IR, as measured by the MCAi, and cancer mortality, but not for the other IR surrogates.

Our finding reinforces the contribution of IR on the pathophysiology of cancer, exemplified by the 40–50% increased risk for cancer related mortality and specifically in individuals with prediabetes.

A number of previous studies have reported an association between increased fasting glucose plasma levels with cancer mortality [14–16, 31, 32]. Parekh et al. [16] demonstrated, as part of the Third National Health and Nutrition Examination Survey (NHANES III; 1988–1994), with an average follow-up of 8.5 years, that the risk for overall cancer mortality was significant higher for every 50 mg/dl increase in fasting plasma glucose concentrations

(HR = 1.22; 95% CI: 1.06–1.39). Our findings show a statistically non-significant 10% increased risk for cancer death in individuals in the upper quartile of fasting glucose, in line with the Parekh et al. findings. Other studies found hyperinsulinemia to relate with increased risk for cancer incidence and mortality (either by a direct mechanism or by interactions with other hormones such as IGF-1) [16, 33–35].

We previously showed [34] after a 29-year follow up of the GOH cohort, that individuals in the upper quartile of the fasting insulin had an increased risk, although with borderline statistical significance, for all-site cancer mortality (HR = 1.37, 95% CI: 0.94, 2.00, p = 0.097). The current analysis showed a 20% non-significant increased risk for the upper fasting insulin quartile as compared to the lower quartiles and a significant 50% higher risk in the upper fasting insulin quartile in pre-diabetics only.

The role of ISIs as predictors for death in cancer patients remain unestablished, with inconsistent findings [16, 35]. Perseghin et al. [35] showed in the Cremona study on a cohort of 2,074 individuals with 15 years of follow-up, a statistically significant but minor association between abnormal HOMA-IR values and death from cancer (HR = 1.003, 95% CI 1.002–1.005, P < 0.001). Our findings support these results as demonstrated by an increased risk for cancer death among individuals in the IR quartiles of the Ln HOMA-IR in individuals with prediabetes (SHR = 1.5, 95% CI: 1.0–2.2). Other studies did not show such a significant association [16].

The MCAi was the only IR surrogate that demonstrated a significant association in the total cohort and exhibited the highest risk for cancer mortality compared with other IR surrogates. In addition, fasting triglycerides were not associated with increased cancer mortality. Each ISI reflects a unique metabolic pathway and evaluate different mechanisms in different stages of insulin resistance [28]. For example, HOMA reflects the interaction between insulin secretion and hepatic glucose production while MCAi further evaluate the impact of insulin resistance on lipid metabolism [19, 21, 28]. Hence, our findings emphasize the importance of elevated fasting triglycerides combined with increase fasting insulin levels, as implemented by MCAi, on cancer prognosis. A possible link between increased triglycerides and cancer incidence was demonstrated in a number of studies [36, 37] through common lipid metabolism pathways (e.g. Malonyl-CoA synthesis) in oncogenesis and adipogenesis [38]. Such a positive association with increased cancer mortality was not observed [39–41] and even correlated with better disease-free survival [42] in breast cancer patients.

In addition, as demonstrated in previous studies [43], MCAi showed the strongest association with insulin resistance, in terms of specificity as well as positive predictive value for distinguishing individuals with metabolic syndrome from healthy adults, compared with other ISIs. Therefore, a combined evaluation of both triglycerides and insulin levels may serve as a more sensitive biomarker for early metabolic syndrome in adults free of diabetes, with a higher risk for cancer mortality compared to other ISIs and each surrogate alone. Further investigation is needed in order to establish triglycerides inter-relationship with cancer progression and prognosis.

In the GOH cohort, an increased risk for all-cause mortality was found in individuals in the IR quartiles of the MCAi, the QUICKI and the HOMA-IR [28]. However, the MCAi was the only ISI that showed a significant association with cardiovascular mortality, regardless of the presence of diabetes. The current findings suggest that the MCAi may be used as a surrogate biomarker for the long-term increased risk of death for both malignancy and cardiovascular morbidities.

The current study did not include participants with the diagnosis of diabetes due to the potential confounding effect of glucose lowering medications [39, 41], as well as the established association between diabetes and cancer mortality [15, 16]. For example, medications for the

treatment of diabetes such as Metformin were negatively associated with mortality among diabetic patient [39] while exogenous insulin use and Sulfonylureas were associated with an increased risk for cancer mortality [44]. However these findings are controversial, due to potential methodological flaws [45]. In addition, diabetic patients display distinct characteristics such as relatively low levels of endogenous insulin as part of the disease progression, and higher BMI, which may confound the association. As previously mentioned, an association between hyperinsulinemia and increased risk for cancer death was observed in a number of studies and thus, lower levels of insulin could potentially have a protective effect from cancer mortality [34]. Moreover, studies have shown better outcome for obese cancer patients, suggesting that increased BMI may serve as good prognostic marker [46].

### 4.1 Strengths and weaknesses of the study

While the standard oral glucose tolerance test (OGTT), as recommended by the American Diabetes Association [47], require an oral administration of 75 gr glucose, in the current study the test was carried out using 100 gr of glucose. This was done due to the absence of clear guidelines at the time of the examination (1979–1982). Furthermore, the ingestion of 100 gr of glucose has been shown to improve insulin sensitivity and insulin secretion with minimal effect on the results of the OGTT in terms of the plasma glucose levels measured throughout the test [48].

In addition, the Yemenite population was over sampled in the GOH cohort beyond its normal proportion in the general Israeli population. This was done in order to increase the statistical power and examine cardiovascular risk factor in this ethnic minority. The multivariable analysis adjusted for ethnicity to overcome this potential confounding.

While the euglycemic insulin clamp is still considered the gold standard for quantifying insulin resistance, in this study we examined other, less invasive and more practical ISIs, which were previously reported to correlate with it well [18–21].

Furthermore, no information on medication, socioeconomic variables or family history were collected during the late 70's intakes. However, the cohort was mainly composed of healthy and employed subjects. In addition, routine screenings for cancer were not widely used at that time. Moreover, during the late 70's, socioeconomic status such as education, rural vs. urban resident etc. were closely correlate with ethnicity in Israel.

Additionally, despite the widely use of ISIs in epidemiological studies, clinical implementation has several limitations such as the absence of general cut-off values and the need for population specific validation (i.e. cut off values may differ according to sex, BMI and ethnicity) [49]. However, due to its feasibility, its low cost, and simplicity, MCAi is superior to other biomarkers and may be implemented in the clinical setting after proper validation.

Finally, the current study did not investigate the association between IR surrogates and cancer site-specific mortality due to the small number of subjects per group.

The study however presents some clear advantages such as the long follow-up over approximately 40 years, the equal representation of both men and women in addition to the representation of an ethnically diverse population. Moreover, blood tests were drawn in the healthy state for research purposes only and analyzed by a single laboratory, avoiding variability in the blood tests analysis. Furthermore, the statistical analysis was performed by two different approaches with similar findings, reinforcing the study results.

## 5. Conclusion

Greater 40-year cancer mortality was observed among adult men and women who were free of diabetes at baseline, but showed higher insulin resistance according to the MCAi. The MCAi should be further studied as an early biomarker for cancer risk in healthy adults.

## Supporting information

**S1 Table. Distribution of malignancy attributed causes of death.**
(PDF)

## Author Contributions

**Conceptualization:** David Rott, Rachel Dankner.

**Data curation:** Yonatan Moshkovits, Angela Chetrit.

**Formal analysis:** Yonatan Moshkovits.

**Investigation:** Yonatan Moshkovits.

**Methodology:** Angela Chetrit, Rachel Dankner.

**Resources:** Angela Chetrit, Rachel Dankner.

**Supervision:** David Rott, Angela Chetrit, Rachel Dankner.

**Validation:** Yonatan Moshkovits, Angela Chetrit, Rachel Dankner.

**Writing – original draft:** Yonatan Moshkovits.

**Writing – review & editing:** David Rott, Rachel Dankner.

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
