## [Decision Letter · Decision Letter 0]

14 Jun 2022

PONE-D-22-11959The insulin sensitivity Mcauley index (MCAi) is associated with 40-year cancer mortality in a cohort of men and women free of diabetes at baselinePLOS ONE

Dear Dr. Dankner,

Thank you for submitting your manuscript to PLOS ONE. After careful consideration, we feel that it has merit but does not fully meet PLOS ONE’s publication criteria as it currently stands. Therefore, we invite you to submit a revised version of the manuscript that addresses the points raised during the review process.

We look forward to receiving your revised manuscript.

Kind regards,

Antonio De Vincentis

Academic Editor

PLOS ONE

Journal Requirements:

Reviewers' comments:

Reviewer's Responses to Questions

**Comments to the Author**

1. Is the manuscript technically sound, and do the data support the conclusions?

Reviewer #1: Yes

Reviewer #2: Partly

2. Has the statistical analysis been performed appropriately and rigorously? 

Reviewer #1: Yes

Reviewer #2: Yes

3. Have the authors made all data underlying the findings in their manuscript fully available?

Reviewer #1: Yes

Reviewer #2: Yes

4. Is the manuscript presented in an intelligible fashion and written in standard English?

Reviewer #1: Yes

Reviewer #2: Yes

5. Review Comments to the Author

Reviewer #1: The authors explode the association between insulin resistance and cancer mortality. The topic is important from both scientific (relating to cancer biology) and clinical (relating to prevention) points of view. A major strength of the study is the long follow-up period of a relatively large cohort. The authors have implemented an interesting and large selection of statistical approaches to analyze the data. The text is well structured, the methods are described extensively and the results are presented clearly.

I have however a couple of questions which need to be addressed in the discussion of the manuscript.

The observed association between MCAi and cancer mortality is interesting, though the lack of such an association with the other IR/IS indices needs some more elaboration. What discriminates MCAi from the other IRIs? Is it possible that a spurious association has been observed?

Please, do some language proof reading.

Some minor remarks:

Table 1: What is the basis for the definition of intermediate BP and hypertension?

Page 18: chapter 3.2: Two distinct statements are mixed in the first sentence. Please re-write.

Reviewer #2: The authors conduct a cohort study of 1,612 Israeli men and women who have been followed for 40 years or until died from a cancer. They investigated the possible association between insulin resistance and the cancer mortality. The authors relay on 4 insulin sensitivity indices to establish the insulin resistance. I would like to congratulate the authors for completing this study and for formulating the research question. The research question is important for the endocrinologist as well as oncology scientist as this question so far is not yet had an explicit answer. The study design and procedure is well conducted yet the inference from the current study is subjected to fulfilling some shortcoming.

The first paragraph discussed epidemiology of cancer in US specifically, which is misleading. I Suggest deleting and keep epidemiological data from Israel.

Since this is a cohort study an efforts should be taken to minimize the potential risk of bias (selection, attrition) to do a very clear inclusion and exclusion criteria is needed .

Although the sample size is sound reasonable, yet a justifiable sample size calculation with sampling technique is needed.

The authors mentions in the limitations that there is no routine investigation for cancer, this is completely understood, if there is any effort done to exclude malignancies should be mentioned.

Why MCAi only independently associated with an increased risk for cancer? The answer should be discussed with emphasis on the TAG which is just mentioned in the discussion.

The reason why the authors has used other indices of insulin resistance is not clear and need to be specify.

At this stage do you think, Are there any clinical implications for this findings?

Minor comments

Please write the p-value in 3 decimal points throughout the manuscript.

Pay careful attention to the coma usage.

6. PLOS authors have the option to publish the peer review history of their article (what does this mean?). If published, this will include your full peer review and any attached files.

Reviewer #1: **Yes: **Alexander Shinkov MD, PhD

Reviewer #2: No

---

## [Author Response · Author response to Decision Letter 0]

5 Jul 2022

Response to reviewers:

Reviewer #1:

The authors explode the association between insulin resistance and cancer mortality. The topic is important from both scientific (relating to cancer biology) and clinical (relating to prevention) points of view. A major strength of the study is the long follow-up period of a relatively large cohort. The authors have implemented an interesting and large selection of statistical approaches to analyze the data. The text is well structured, the methods are described extensively and the results are presented clearly.

Response: Thank you for your positive comment.

The observed association between MCAi and cancer mortality is interesting, though the lack of such an association with the other IR/IS indices needs some more elaboration. What discriminates MCAi from the other IRIs? Is it possible that a spurious association has been observed.

Response: Thank you for your comment. We have elaborated in the discussion section on the subject emphasizing the uniqueness of MCAi as the only ISI that evaluate the impact of insulin resistance on lipid metabolism and as a more sensitive biomarker for insulin resistance:

“Each ISI reflects a unique metabolic pathway and evaluate different mechanisms in different stages of insulin resistance. For example, HOMA reflects the interaction between insulin secretion and hepatic glucose production while MCAi further evaluate the impact of insulin resistance on lipid metabolism. Hence, our findings emphasize the importance of elevated fasting triglycerides combined with increase fasting insulin levels, as implemented by MCAi, on cancer prognosis.”

“In addition, as demonstrated in previous studies, MCAi showed the strongest association with insulin resistance, in terms of specificity as well as positive predictive value, for distinguishing individuals with metabolic syndrome from healthy adults, compared with other ISIs. Therefore, a combined evaluation of both triglycerides and insulin levels may serve as a more sensitive biomarker for early metabolic syndrome in adults free of diabetes with a higher risk for cancer mortality compared to other ISIs and each surrogate alone.”

Table 1: What is the basis for the definition of intermediate BP and hypertension?

Response: Thank you for your comment. Following this comment, and in order to increase its informativeness, blood pressure is now analyzed as a continuous variable. As systolic and diastolic blood pressure highly correlated (r=0.76) only systolic blood pressure entered the final regression model.

Page 18: chapter 3.2: Two distinct statements are mixed in the first sentence. Please re-write.

Response: Thank you for your comment. The paragraph was re-edited as follow:

“During a mean follow up time of 36.7±0.2 years, 970 (60.2%) participants died and 47,191 person years were accrued. Cancer was the second most common cause of death with 264 (16.4%) deaths attributed to cancer related mortality.”

 

Reviewer #2:

The authors conduct a cohort study of 1,612 Israeli men and women who have been followed for 40 years or until died from a cancer. They investigated the possible association between insulin resistance and the cancer mortality. The authors relay on 4 insulin sensitivity indices to establish the insulin resistance. I would like to congratulate the authors for completing this study and for formulating the research question. The research question is important for the endocrinologist as well as oncology scientist as this question so far is not yet had an explicit answer. The study design and procedure is well conducted yet the inference from the current study is subjected to fulfilling some shortcomings.

Response: Thank you for your positive feedback.

The first paragraph discussed epidemiology of cancer in US specifically, which is misleading. I Suggest deleting and keep epidemiological data from Israel.

Response: Thank you for your comment. The paragraph on cancer epidemiology in the US was significantly shortened to emphasize the epidemiological data from Israel.

Since this is a cohort study an efforts should be taken to minimize the potential risk of bias (selection, attrition) to do a very clear inclusion and exclusion criteria is needed.

Response: Thank you for your comment. As mentioned in the method section, participants were randomly drawn from the national population registry according to strata of sex, country of birth, and birth decade.

In addition, we further elaborated on the inclusion and exclusion criteria as follow (pages 7-8):“Inclusion criteria for the current study included the absence of diabetes at baseline and the availability of data on both fasting glucose and insulin plasma levels at baseline. Individuals who died from cancer within the first 2 years of follow-up were excluded from the cohort. Out of 2769 participants primarily examined in the second phase, 1,612 met the inclusion criteria”.

Moreover, in order to avoid selection-bias, we compared baseline characteristics such as age, sex, ethnicity, blood pressure etc. between the original cohort and the final sample with similar distribution, as discussed in the method section (page 5). 

Finally, the study is practically free of attrition, as information on the outcome, i.e. death, was achieved by linkage of the study file with the Israeli population registry. The likelihood that individuals from the cohort have migrated from Israel to other countries is extremely low. A previous study showed 1.7% of Israeli citizens from similar age ranges as in our study were lost to follow-up due to migration.

Although the sample size is sound reasonable, yet a justifiable sample size calculation with sampling technique is needed.

Response: Thank you for your comment. Sample size calculation was added to the method section (page 7) as follow:

Sample size was calculated using WINPEPI software implementing the Z test for proportion analysis with 80% power and 5% significance level. Based on previous publication on the study cohort, assuming an average probability of survival to end of follow-up of 35% for individuals at the higher quartiles of the ISIs and a minimal Hazard ratio of 1.2, the calculated total sample size was 1,548 subjects.” 

The authors mentions in the limitations that there is no routine investigation for cancer, this is completely understood, if there is any effort done to exclude malignancies should be mentioned.

Response: Thank you for your comment. Unfortunately, data regarding the presence of cancer at baseline was not available. However, participants who died from cancer related mortality during the first two years of follow-up were excluded in order to mitigate reverse causation.

Why MCAi only independently associated with an increased risk for cancer? The answer should be discussed with emphasis on the TAG which is just mentioned in the discussion.

Response: Thank you for your comment. We have elaborated in the discussion section on the subject emphasizing the uniqueness of MCAi as the only ISI that evaluate the impact of insulin resistance on lipid metabolism and as a more sensitive biomarker for insulin resistance:

“Each ISI reflects a unique metabolic pathway and evaluate different mechanisms in different stages of insulin resistance. For example, HOMA reflects the interaction between insulin secretion and hepatic glucose production while MCAi further evaluate the impact of insulin resistance on lipid metabolism. Hence, our findings emphasize the importance of elevated fasting triglycerides combined with increase fasting insulin levels, as implemented by MCAi, on cancer prognosis.”

“In addition, as demonstrated in previous studies, MCAi showed the strongest association with insulin resistance, in terms of specificity as well as positive predictive value, for distinguishing individuals with metabolic syndrome from healthy adults, compared with other ISIs. Therefore, a combined evaluation of both triglycerides and insulin levels may serve as a more sensitive biomarker for early metabolic syndrome in adults free of diabetes with a higher risk for cancer mortality compared to other ISIs and each surrogate alone.”

The reason why the authors has used other indices of insulin resistance is not clear and need to be specify.

Response: Thank you for your comment. The main objective of the study was to evaluate several insulin resistance surrogates, such as plasma glucose and insulin levels as well as insulin sensitivity indices, and their association with cancer mortality. Each marker represents a unique mechanism of insulin resistance in different stages of the disease. For example, HOMA reflects the interaction between insulin secretion and hepatic glucose production while MISI evaluate the total body insulin resistance levels.

We now elaborate more on the subject in the discussions section:

“Each ISI reflects a unique metabolic pathway and evaluate different mechanisms in different stages of insulin resistance. For example, HOMA reflects the interaction between insulin secretion and hepatic glucose production while MCAi further evaluate the impact of insulin resistance on lipid metabolism. Hence, our findings emphasize the importance of elevated fasting triglycerides combined with increase fasting insulin levels, as implemented by MCAi, on cancer prognosis.”

At this stage do you think, Are there any clinical implications for this findings?

Response: Thank you for your comment. We further elaborated on the subject under the limitations section:

“In addition, despite the widely use of ISIs in epidemiological studies, clinical implementation has several limitations such as the absence of general cut-off values and the need for population specific validation (i.e. cut off values may differ according to sex, BMI and ethnicity). However, due to its feasibility, its low cost, and simplicity, MCAi seems superior to other biomarkers and could be implemented in the clinical setting after proper validation”.

Please write the p-value in 3 decimal points throughout the manuscript

Response: Thank you for your comment. The p-value is now written accordingly.

respond to specific editor comments : all comments were addressed

---

## [Decision Letter · Decision Letter 1]

20 Jul 2022

The insulin sensitivity Mcauley index (MCAi) is associated with 40-year cancer mortality in a cohort of men and women free of diabetes at baseline

PONE-D-22-11959R1

Dear Dr. Dankner,

We’re pleased to inform you that your manuscript has been judged scientifically suitable for publication and will be formally accepted for publication once it meets all outstanding technical requirements.

Kind regards,

Antonio De Vincentis

Academic Editor

PLOS ONE

Additional Editor Comments (optional):

Reviewers' comments:

Reviewer's Responses to Questions

**Comments to the Author**

1. If the authors have adequately addressed your comments raised in a previous round of review and you feel that this manuscript is now acceptable for publication, you may indicate that here to bypass the “Comments to the Author” section, enter your conflict of interest statement in the “Confidential to Editor” section, and submit your "Accept" recommendation.

Reviewer #1: All comments have been addressed

Reviewer #2: All comments have been addressed

2. Is the manuscript technically sound, and do the data support the conclusions?

Reviewer #1: Yes

Reviewer #2: Yes

3. Has the statistical analysis been performed appropriately and rigorously? 

Reviewer #1: Yes

Reviewer #2: Yes

4. Have the authors made all data underlying the findings in their manuscript fully available?

Reviewer #1: Yes

Reviewer #2: Yes

5. Is the manuscript presented in an intelligible fashion and written in standard English?

Reviewer #1: Yes

Reviewer #2: Yes

6. Review Comments to the Author

Reviewer #1: The authors have answered all the questions and made the recommended changes. I do not have any further comments.

Reviewer #2: Thank you for addressing the previous comments. The manuscript is much improved. I have no further comments

7. PLOS authors have the option to publish the peer review history of their article (what does this mean?). If published, this will include your full peer review and any attached files.

Reviewer #1: No

Reviewer #2: **Yes: **HZ Hamdan

---

## [Editor Report · Acceptance letter]

26 Jul 2022

PONE-D-22-11959R1 

The insulin sensitivity Mcauley index (MCAi) is associated with 40-year cancer mortality in a cohort of men and women free of diabetes at baseline 

Dear Dr. Dankner:

I'm pleased to inform you that your manuscript has been deemed suitable for publication in PLOS ONE. Congratulations! Your manuscript is now with our production department. 

Kind regards, 

on behalf of

Dr. Antonio De Vincentis 

Academic Editor

PLOS ONE